green chemistry/organic chemistry/synthetic chemistry

regioselectivity, ultrasonic irradiation, bis-[1,2,4]-triazol-3-yl amines, bis-2-iminothiazolines, green metrics

**Author for correspondence:**
Wael Abdelgayed Ahmed Arafa
e-mail: waa00@fayoum.edu.eg

This article has been edited by the Royal Society of Chemistry, including the commissioning, peer review process and editorial aspects up to the point of acceptance.

# Sustainable and scalable synthesis of polysubstituted bis-1,2,4-triazoles, bis-2-iminothiazolines and bis-thiobarbiturates using bis-N,N-disubstituted thioureas as versatile substrate

Wael Abdelgayed Ahmed Arafa[1,2]
and Hamada Mohamed Ibrahim[2]

[1]Department of Chemistry, College of Science, Jouf University, PO Box 2014, Sakaka, Aljouf, Kingdom of Saudi Arabia
[2]Department of Chemistry, Faculty of Science, Fayoum University, PO Box 63514 Fayoum City, Egypt

WAAA, 0000-0002-9288-4143

An expedient and tandem regioselective one-pot protocol for the sono-synthesis of bis-[1,2,4]-triazol-3-yl amines and bis-2-iminothiazolines from corresponding bis-1,3-disubstituted thioureas has been developed. The products' regioselectivity correlate well with the $pK_a$s of the parent amines, in which the amine possessing higher $pK_a$ goes to the ring nitrogen, whereas the other nitrogen remains flanked as an exocyclic nitrogen of the bis-triazole or bis-thiazole moieties. Further, the sonochemical preparation of both bis-5-(2-nitrobenzylidene) thiobarbiturates and bis-2-thioxoimidazolidine-4,5-diones from bis-1,3-disubstituted thioureas has also been achieved. The obtained bis-5-(2-nitrobenzylidene)thiobarbiturates easily underwent reductive cyclization to afford the corresponding bis-5-benzo[c]isoxazol-3-ylidenethiobarbiturates. The scope and limitations of these strategies have been studied. Moreover, the suggested methodologies have advantages such as broad functional group tolerance, mild conditions, operational simplicity and applicability on a gram scale. Furthermore, the protocols scored well in a number of green metrics, subsequently showing these approaches to be environmentally benign and sustainable processes.

# 1. Introduction

As of late, organic chemists have given careful consideration to heterocyclic compounds in light of their potential significance in the agricultural and pharmaceutical fields. 1,2,4-Triazole and bis-1,2,4-triazole derivatives have been well reported to show wide potential applications in chemical, medicinal, supramolecular, agricultural and also in materials sciences [1–6]. Furthermore, a considerable number of triazole-based medicines have been broadly used in clinic, for example, Letrozole [2], Voriconazole [7] and Anastrozole [8], which have been used as non-steroidal aromatase inhibitors in drugs for breast cancer treatment. In spite of the numerous protocols reported in the literature for the preparation of [1,2,4]-triazoles having diversified substituents, few of these mention the preparation of the triazole possessing no substitution at position 5 [6]. Furthermore, most of these methods suffer from serious ecological concerns as they comprise the utilization of strong acidic or basic conditions, expensive and poisonous chemicals, thereby impeding the application of these protocols. Hence, development of more sustainable methodology for the assembly of bis-[1,2,4]-triazoles is therefore of great significance. Thiazoles and their bis-heterocycles stand out among the most important moieties in drug design and heterocyclic chemistry. Thiazole motif is broadly found in some naturally occurring substances and varied pharmaceutically active compounds [9–13]. As well, thiobarbituric acids (TBA) and their 5-arylidine derivatives have turned out to be attractive to medicinal chemists because of their broad scope of biological activities and as intermediates in the preparation of heterocyclic compounds [14–19]. Also, thiobarbiturates are broadly used for the synthesis of diverse complexes [20,21], being employed as a reagent for spectrophotometric investigation of numerous metals [20–23], to determine the oxidation degree of natural fats in nutrients [24–26] and industrially as thermal stabilizers for rigid poly(vinyl chloride) at high temperature [27]. Imidazole motif exists widely in different types of naturally occurring and synthetic pharmaceutical compounds and displays assortment of pharmacological activities [28–30]. Moreover, imidazole derivatives are used as building blocks in natural product synthesis [31], catalysis [32] and coordination chemistry [33]. In the novel effective and facile synthetic protocols of heterocycles, the merits of a strategy are measured by speed, inexpensive starting materials and energy sources. In this regard, utilization of sonochemistry as attractive methodologies to realize these aims has been recently reported [34]. Small cavities that generated as a result of ultrasonic waves travel through the liquid and lead to chemical reactions. This cavitation leads to the creation of micro-bubbles in the liquid, which on imploding generate high temperatures and pressures in their surroundings [35]. In comparison with traditional procedures, decreased reaction times, improved yields and diminished energy used are significant merits of ultrasound methodology. As a part of our ongoing research plan on the preparation of diversified biologically active bis-heterocyclic compounds and on interesting new and eco-friendly green synthetic protocols [36–41], a straightforward synthetic procedure for the synthesis of novel series of bis-heterocycles can be effectively accomplished under ultrasound irradiation as a viable greener technique. Furthermore, the electronic impact of the 1,3-substituents on the bis-thiourea performs an essential role on the regioselectivity of the cyclization and product distribution.

# 2. Results and discussion

The genesis of this work has commenced with one of our recent papers on a green synthetic procedure for the synthesis of asymmetrically substituted bis-thioureas (scheme 1) [39]. The bis-thioureas (**3a–d**) were obtained in excellent yields by the reaction of isothiocyanates (**1a, b**) with diamines (**2a, b**) in $CH_3CN$ by using ultrasound irradiation as a sustainable energy source (scheme 1).

Firstly, we thought of developing an eco-friendly one-pot protocol for the assembly of bis-3-amino-[1,2,4]-triazoles. Our aim was to select chemicals and optimize different reaction conditions to find a general protocol that can be used to synthesize regioselective bis-[1,2,4]-triazol-3-yl amines via oxidative desulfurization reaction. The feasibility of our envisioned protocol can be tested by conducting the following model reaction: to a solution of bis-thiourea **3a** (1.0 equiv) in ethanol (10 ml) were added 2.0 equiv of formic hydrazide and 2.0 equiv of $HgCl_2$ portion-wise over a time of 5 min at ambient temperature (25°C). Generation of a novel product alongside other little side products was observed together with the retention of some starting materials. Purification and characterization by usual methods revealed the product to be hitherto unreported 4,4′-[(1R,4R)-cyclohexane-1,4-diyl)bis(N-benzyl-4H-1,2,4-triazol-3-amine] (**4a**) (17% isolated yield, table 1, entry 1). Our next target was to identify an appropriate thiophile to accomplish this reaction. We selected to compare two mercury(II)

**Scheme 1.** An ultrasound-assisted methodology for the preparation of unsymmetrically substituted bis-thioureas **3a–d** [39].

**Table 1.** Optimization of reaction conditions for preparation of **4a**.

| entry | thiophile | method | thiophile molar ratio | time (min) | yield (%) |
|---|---|---|---|---|---|
| 1 | HgCl$_2$ | stirring | 2.0 | 60 | 17 |
| 2 | Hg(OAc)$_2$ | stirring | 2.0 | 60 | 22 |
| 3 | I$_2$/K$_2$CO$_3$ | stirring | 2.0 | 60 | 67 |
| 4 | I$_2$/K$_2$CO$_3$ | US | 2.0 | 15 | 90 |
| 5 | I$_2$/K$_2$CO$_3$ | US | 2.0 | 10 | 86 |
| 6 | I$_2$/K$_2$CO$_3$ | US | 2.0 | 20 | 90 |
| 7 | I$_2$/K$_2$CO$_3$ | US | 2.0 | 40 | 90 |
| 8 | I$_2$/K$_2$CO$_3$ | US | 3.0 | 15 | 96 |
| 9 | I$_2$/K$_2$CO$_3$ | US | 4.0 | 15 | 96 |

salts (table 1, entries 1 and 2) and the I$_2$/K$_2$CO$_3$ reagent (table 1, entry 3) as thiophile and checked the generation of **4a** by thin layer chromatography (TLC) in the model reaction.

According to the yields of the product (table 1, entries 1–3), it was obvious that I$_2$/K$_2$CO$_3$ was the optimal thiophile for this model reaction, and henceforth we chose to use I$_2$/K$_2$CO$_3$ for all next reactions as the thiophile. Interestingly, when the aforesaid model reaction was conducted under ultrasound (US) irradiation, the required derivative (**4a**) acquired in good yield (90%) and in short reaction time (table 1, entry 4). Screening of some other inorganic salts, for example, KHCO$_3$, Na$_2$CO$_3$, NaHCO$_3$ and Cs$_2$CO$_3$ were also found to be evenly efficacious. In a further reaction, it was also observed that shorter periods of sonication afforded diminished yields (table 1, entry 5), whereas longer periods gave no advantage (table 1, entries 6 and 7). By varying the amount of thiophile in the above reaction (table 1, entries 8 and 9), it was found that 3.0 equiv of I$_2$/K$_2$CO$_3$ provided the best yield of **4a** in 96% (table 1, entry 8). Finally, the optimized conditions for this desulfurization reaction involved sonication of 1.0 equiv of **3a** and 2.0 equiv of NH$_2$NHCHO with 3.0 equiv of I$_2$/K$_2$CO$_3$ in ethanol (10 ml) for 15 min at ambient temperature which offered an excellent 96% yield of **4a** (table 1,

**Table 2.** Substrate scope for bis-[1,2,4]triazoles **4** and **5**.

$R = C_6H_5, CH=CH_2$; R' =

|       |          |                      | yield (%)     |               |
| ----- | -------- | -------------------- | ------------- | ------------- |
| entry | R        | R'                   | 4             | 5             |
| 1     | $C_6H_5$ | cyclohexyl           | 93 (**4a**)   | 0             |
| 2     | $CH=CH_2$ | cyclohexyl          | 90 (**4b**)   | 0             |
| 3     | $C_6H_5$ | $-CH_2-C_6H_4-CH_2-$ | 57 (**4c**)   | 43 (**5a**)   |
| 4     | $CH=CH_2$ | $-CH_2-C_6H_4-CH_2-$ | 50 (**4d**)   | 50 (**5b**)   |

entry 8). Because our target is to develop a more environmentally friendly methodology, non-environmental organic bases and solvents have not been studied for this conversion.

Since the formation of bis-[1,2,4]triazole (**4a**) is an addition-dehydration reaction, we postulated that the substituents linked to the thiourea nitrogens would perform an essential role in the ring closure regioselectivity. To investigate the impact of substituent on the construction of the bis-[1,2,4]triazoles, a series of asymmetrical bis-thioureas (**3a–d**) were studied. A one-pot reaction was accomplished with the bis-thioureas (**3a–d**), $I_2$, $K_2CO_3$, and $NH_2NHCHO$ (1:3:3:2 equiv) blended together at ambient temperature with sonication for 15 min. The outcomes are outlined in table 2.

From the results outlined in table 2, a good correlation between $pK_a$ of the parent amines and product regioselectivity could be set up. According to the suggested mechanism in scheme 2, the intermediate (**I**), that formed under the influence of catalyst, underwent desulfurization reaction to afford intermediate (**II**). Then, the formic hydrazide attacked the *in situ* produced unsymmetrical bis-carbodiimide (**II**) followed by protonation at the nitrogen having higher $pK_a$ (scheme 2) to afford intermediate (**III**) instead of at the nitrogen having lower $pK_a$. After that, intermediate (**III**) underwent an intramolecular nucleophilic attack from nitrogen lone pair onto the aldehydic carbonyl group to give structure (**IV**). Finally, intermediate (**IV**) underwent a dehydration step to furnish the regioisomeric bis-[1,2,4]-triazole derivative **4a**. Therefore, the nitrogen atom of amine linked to the thiourea possessing lower $pK_a$ stays as exocyclic nitrogen in the final product, whereas the other nitrogen atom (having higher $pK_a$) would be involved in the ring. Also, this has been confirmed during our regioselective preparation of thiazole-2-imines [39]. Accordingly, it is anticipated that large difference in values of $pK_a$ of amines linked to bis-thiourea should afford solely one regioisomer, whereas small difference should give two regioisomers. For example, under the optimized reaction condition, the asymmetrical bis-thiourea **3c** comprising both benzyl amine ($pK_a = 9.34$) and *p*-xylylene diamine ($pK_a = 9.46$) provided a mixture of both regioisomers **4c** and **5a** in the ratio of 57 : 43 (table 2, entry 3). Also, in the case of using the unsymmetrical bis-thiourea **3d** containing allyl amine ($pK_a = 9.49$) and *p*-xylylene diamine ($pK_a = 9.46$), both regioisomers **4d** and **5b** were obtained in the ratio of 50 : 50 (table 2, entry 4). Whereas, other thioureas (**3a, b**) possessing a $pK_a$ difference of 1.46 and 1.34 units, respectively, affording only one regioisomer (**4a** and **4b**, respectively; table 2, entries 1 and 2). Notwithstanding the isolation of water during the reaction (scheme 2), no traces of urea (as a result of water attack to carbodiimide intermediate (**II**)) were detected; probably attributable to the more nucleophilic nature of the formic acid hydrazide compared with water.

The conventional preparation of thiazoles includes the Hantzsch condensation reaction of α-haloketones and thioureas, which affords after dehydrohalogenation the 2-iminothiazoles [42]. In the reinvestigation of the synthesis of thiazoles using Hantzsch methods under sonication, six new bis-2-iminothiazolines **7–9**

**Scheme 2.** A postulated mechanism for the preparation of bis-[1,2,4]triazoles **4a**.

were acquired in excellent yields. Thus, reaction of bis-thioureas **3a−d** with *o*-methoxy phenacyl bromide (**6**) was carried out using catalytic amount of sodium acetate and sonication at 80°C for 10 min to give bis-2-iminothiazolines **7−9** (scheme 3). Unsymmetrical thioureas **3a**, **b** (with higher p$K_a$ differences) smoothly underwent the reaction to give the sole regioisomeric products **7a**, **b** in a slightly preferable yield than those **3c**, **d** (with lesser p$K_a$ differences) which afforded a mixture of regioisomeric products **8** and **9** (scheme 3).

Thereafter, on heating (60°C) of derivatives **3a−d** (1 mmol) with malonic acid (2.2 mmol) in acetyl chloride (6.0 mmol) or oxaloyl chloride (2.2 mmol) afforded the hitherto unreported bis-thiobarbituric acids (**10a−d**) and bis-imidazoles (**11a−d**), respectively, with inadmissible results (scheme 4). Consequently, some reaction variables, for instance, reagents stoichiometry, energy sources and times, were investigated using bis-thiourea **3a** as the model substrate for improving reaction yield. Optimization resulted in 1.0 equiv of **3a**, 2.2 equiv of oxaloyl chloride or 2.2 equiv malonic acid (in 6.0 equiv of acetyl chloride) under sonication at 40°C for 7 min, affording derivatives **10a** and **11a** in 99% and 96%, respectively. After establishing an effective protocol for these reactions, a diversified range of bis-thioureas (**3a−d**) were employed to synthesize functionalized bis-thiobarbituric acids (**10a−d**) and bis-imidazoles (**11a−d**). It was noticed that the electronic nature of bis-thioureas had a minor impact on the reaction effectiveness, afforded the corresponding products in excellent yields. Different from derivatives **3a−d**, FT-IR and $^{13}$C NMR spectra of derivatives **10** and **11** exhibited additional signals originated from carbonyl functions at the related regions. Furthermore, in the $^1$H NMR spectra of derivatives **10** and **11**, no signals were recorded corresponding to the NH groups (thiourea moiety).

2,1-Benzisoxazole (anthranil) is a privileged heterocycle that is used for the synthesis of biologically active derivatives such as acridines and quinolones [43,44]. To best of our knowledge, bis-2,1-benzisoxazoles possessing *exo*-double bonds at 5-position have not been reported. We herein present the first synthesis of bis-2,1-benzisoxazoles (**13a−d**) from bis-thiobarbituric acids (**10a−d**). In a prototypical reaction, to a solution of bis-thiobarbituric acids (**10a−d**) (1.0 mmol) in water (10 ml), was added *o*-nitrobenzaldehyde (2.1 mmol) and the obtained mixture was sonicated (40 MHz) at 80°C for 5 min, which afforded the desired products (**12a−d**) in excellent yields (scheme 5). Then, the preparation of bis-2,1-benzisoxazoles (**13a−d**) was accomplished from the reduction of derivatives (**12a−d**) with SnCl$_2$.2H$_2$O/HCl in ethanol under sonication for 10 min at 80°C (scheme 5).

**Scheme 3.** An ultrasound-assisted methodology for the preparation of bis-iminothiazoles **7–9**.

**Scheme 4.** An ultrasound-assisted methodology for the preparation of **10** and **11**.

Interestingly, compounds **13a–d** could be also synthesized via one-pot consecutive reaction. After sonication of the solution of bis-thiobarbituric acids (**10a–d**) (1.0 mmol) and *o*-nitrobenzaldehyde (2.0 mmol) in water (10 ml) for 5 min, water was removed under vacuum and the remaining residue was suspended in ethanol, to which, $SnCl_2.2H_2O/HCl$ were added. The resulting mixture was then sonicated for 10 min at 80°C. The mechanism of this reaction presumably involves generation of the non-separable bis-nitrosoarene intermediate (**VI**), via a partial reduction of the nitro groups, which would subsequently act as a nucleophile able to attack the bis-thiobarbiturate in a typical Michael addition reaction to yield the desired products **13a–d** (scheme 5). The suggested structures (scheme 5) were approved by spectral analyses. In the $^1H$ NMR spectra of derivatives **13a–d**, two doublets were

**Scheme 5.** An ultrasound-assisted methodology for the preparation of **13a – d**.

**Table 3.** Calculated green metrics for the scaled-up preparation of compounds **4a**, **7a**, **10a**, **13a**.

| entry | derivative | yield | | EF | PMI | AE(%) | CE(%) | RME(%) | YE(%) |
|-------|-----------|-------|-----|-----|-----|-------|-------|--------|-------|
| | | g | % | | | | | | |
| 1 | **4a** | 4.2 | 98 | 0.019 | 1.3 | 91 | 100 | 79 | 6.5 |
| 2 | **7a** | 6.5 | 96 | 0.038 | 1.3 | 78 | 100 | 74 | 9.6 |
| 3 | **10a** | 5.3 | 98 | 0.14 | 1.2 | 88 | 100 | 86 | 14 |
| 4 | **13a** | 7.6 | 97 | 0.03 | 1.1 | 92 | 100 | 89 | 6.5 |

observed in the ranges of 7.98–7.80 and 6.80–6.78 ppm attributed to the *o*-substituted benzene ring. Also, in the $^{13}$C NMR spectra, the signals at about 179, 161 and 86 ppm were specified to C=S, C=O and methylenic carbon, respectively.

The enhancement in the rate and yield of these sonochemical reactions may be attributed to the acoustic cavitation [45], in which the vapour bubbles that form in low-pressure regions collapse when subjected to higher pressure, with the implosion generating extremely high localized temperatures more than 5000 K and high pressures more than 1000 atm [46]. Collapsing bubbles filled with diaromatic vapour produce localized points of extremely high temperature 'hotspot' that are sufficient to drive the reactions within short time in high yields [47].

In order to demonstrate the potential for scale-up of the aforementioned protocols, we performed representative reactions on the gram scale and the isolated desired products are summarized in table 3. Some green metrices [48] such as E-factor (EF), process mass intensity (PMI), atom economy (AE), carbon efficiency (CE), reaction mass efficiency (RME) and yield economy (YE) were calculated for these processes and the results are tabulated in table 3. The higher environmental compatibility green metrics such as smaller values of EF and higher values of PMI, AE, CE, RME and YE confirm the environmental friendliness of the current protocols.

# 3. Conclusion

In conclusion, straightforward and highly effective regioselective protocols for the sono-synthesis of bis-[1,2,4]-triazol-3-yl amines and bis-2-iminothiazolines from corresponding asymmetrically substituted bis-thioureas have been successfully demonstrated. The product regioselectivity correlated well with the p$K_a$s of the parent amines; the amine possessing higher p$K_a$ become part of the ring, while that having lower p$K_a$ remains as an exocyclic nitrogen in the final product. Therefore, the present protocol afforded an approach to the generation of one regioisomer through appropriate tuning of the p$K_a$s of the used amines. Furthermore, an efficient methodology to sono-synthesize bis-isoxazolthiobarbiturates and bis-imidazoles from bis-1,3-disubstituted thioureas has been also described. The reactions proceed smoothly affording target products in excellent yields. Moreover, an assortment of green metrics has been discussed and our protocol exemplary fit in this grid.

# 4. Experimental procedure

## 4.1. General information

NMR spectra were determined on Bruker Ultra Shield spectrometer at 400 ($^1$H) and 100 ($^{13}$C) MHz. Chemical shifts are recorded in parts per million (ppm) and are referenced to TMS. Analytical thin layer chromatography (TLC) was employed on a silica gel plate (Merck® 60F254). IR spectra were recorded on Perkin–Elmer Spectrum One spectrometer. Sonication was performed in a SY5200DH-T ultrasound cleaner. Melting points were measured on Electrothermal IA9100 melting point apparatus (UK). All chemicals were commercially available and were used without any purification.

## 4.2. Synthesis of bis-1,2,4-triazol-3-amine derivatives **4 – 5**

To a solution of bis-thioureas **3a–d** (1.0 mmol) in ethanol (10 ml) were added sequentially NH$_2$NHCHO (2.0 mmol) and aqueous K$_2$CO$_3$ (3.0 mmol in 1 ml water) and the mixture was sonicated at ambient temperature (25°C). Then, I$_2$ (3.0 mmol) was appended portion-wise within 5 min. After that, the reaction mixture was sonicated for a further 10 min (monitoring by TLC). During this period, the iodine colour vanished with the appearance of elemental sulfur. After filtration, the solvent was removed under reduced pressure and the remaining residue was treated with 5% sodium thiosulfate (5 ml). The reaction product was then purified over a column of silica gel and eluted with DCM/MeOH mixture to afford the corresponding compounds **4** and **5**.

### 4.2.1. 4,4′-((1R,4R)-Cyclohexane-1,4-diyl)bis(N-benzyl-4H-1,2,4-triazol-3-amine) **4a**

Yellow crystals, yield 96%, m.p. 296–298°C; IR (KBr): $\nu$/cm$^{-1}$ 3328 (NH), 1610 (C=N); $^1$H NMR (400 MHz, DMSO-$d_6$): $\delta$ 8.60 (s, 2H, triazole-H), 7.35–7.25 (m, 10H, Ar-H), 4.13 (s, 4H, CH$_2$), 3.93 (br, 2H, cyclohexyl-H), 1.92–1.91 (m, 4H, cyclohexyl-H), 1.26–1.19 ppm (m, 4H, cyclohexyl-H); $^{13}$C NMR (100 MHz, DMSO-$d_6$): $\delta$ 150.2 (triazole-C3), 142.4 (triazole-C5), 138.4, 129.9, 128.6, 127.6 (Ar-C), 51.4 (cyclohexyl-C), 47.7 (CH$_2$), 30.3 ppm (cyclohexyl-C); MS (EI): m/z (%) 429 (M$^+$+1, 2.2), 428 (M$^+$, 11.6), 337 (M$^+$−91, 100).

### 4.2.2. 4,4′-((1R,4R)-Cyclohexane-1,4-diyl)bis(N-allyl-4H-1,2,4-triazol-3-amine) **4b**

Yellow crystals, yield 95%, m.p. 285–286°C; IR (KBr): $\nu$/cm$^{-1}$ 3310 (NH), 1625 (C=N); $^1$H NMR (400 MHz, DMSO-$d_6$): $\delta$ 8.56 (s, 2H, triazole-H), 5.91–5.80 (m, 2H, −CH$_2$−CH=CH$_2$), 5.17–5.05 (m, 4H, −CH$_2$−CH=CH$_2$), 4.06 (br, 4H, −CH$_2$−CH=CH$_2$), 3.18–3.13 (br, 2H, cyclohexyl-H), 1.78–1.76 (m, 4H, cyclohexyl-H), 1.46–1.40 ppm (m, 4H, cyclohexyl-H); $^{13}$C NMR (100 MHz, DMSO-$d_6$): $\delta$ 151.6 (triazole-C3), 142.9 (triazole-C5), 135.3 (−CH$_2$−CH=CH$_2$), 115.7 (−CH$_2$−CH=CH$_2$), 51.8 (cyclohexyl-C), 45.9 (−CH$_2$−CH=CH$_2$), 30.7 ppm (cyclohexyl-C); MS (EI): m/z (%) 329 (M$^+$+1, 4.6), 328 (M$^+$, 100).

### 4.2.3. 4,4′-(1,4-Phenylenebis(methylene))bis(N-benzyl-4H-1,2,4-triazol-3-amine) **4c** and
N,N′-(1,4-phenylenebis(methylene))bis(4-benzyl-4H-1,2,4-triazol-3-amine) **5a**

The crude reaction mixture comprising both the non-isolable regioisomeric products was purified by column chromatography (96% DCM/MeOH) to afford the mixture of regioisomers **4c** and **5a** in the

ratio of (57 : 43); Rf = 0.37. Yellow crystals, yield 96%, m.p. 256–259°C; IR (KBr): $\nu$/cm$^{-1}$ 3329 (NH), 1609 (C=N); $^1$H NMR (400 MHz, DMSO-$d_6$): $\delta$ 8.674 (s, 2H, triazole-H), 8.670 (s, 2H, triazole-H), 7.32–7.18 (m, Ar-H), 4.84–4.82 (m, C$H_2$), 4.45–4.42 ppm (m, C$H_2$); $^{13}$C NMR (100 MHz, DMSO-$d_6$): $\delta$ 153.0 (triazole-C3), 152.6 (triazole-C3), 145.5 (triazole-C5), 145.3 (triazole-C5), 138.9, 138.6, 137.0, 129.5, 129.2, 128.8, 128.3, 128.1, 127.5, 127.1, 126.8, 126.5 (Ar-C), 47.6, 47.5, 45.8 ppm (C$H_2$); MS (EI): $m/z$ (%) 451 (M$^+$+1, 0.9), 450 (M$^+$, 4.2), 359 (M$^+$−91, 100).

### 4.2.4. 4,4′-(1,4-Phenylenebis(methylene))bis(N-allyl-4H-1,2,4-triazol-3-amine) **4d** and N,N′-(1,4-phenylenebis(methylene))bis(4-allyl-4H-1,2,4-triazol-3-amine) **5b**

The crude reaction mixture comprising both the non-isolable regioisomeric products was purified by column chromatography (98% DCM/MeOH) to give the mixture of regioisomers **4d** and **5b** in the ratio of (50 : 50); Rf = 0.41. Yellow crystals, yield 95%, m.p. 235–239°C; IR (KBr): $\nu$/cm$^{-1}$ 3285 (NH), 1612 (C=N); $^1$H NMR (400 MHz, DMSO-$d_6$): $\delta$ 8.607 (s, 2H, triazole-H), 8.600 (s, 2H, triazole-H), 7.34–7.27 (m, Ar-H), 5.88–5.79 (m, −C$H_2$−CH=CH$_2$), 5.62–5.60 (m, −CH$_2$−C$H$=CH$_2$), 5.17–4.97 (m, −CH$_2$−CH=C$H_2$), 4.75–4.69 (m, −C$H_2$−CH=CH$_2$), 4.26 (br, C$H_2$), 4.03 ppm (s, C$H_2$); $^{13}$C NMR (100 MHz, DMSO-$d_6$): $\delta$ 152.3 (triazole-C3), 152.1 (triazole-C3), 145.3 (triazole-C5), 145.0 (triazole-C5), 137.4, 135.7, 134.3, 128.5, 128.2 (Ar-C), 135.6, 135.3 (−CH$_2$−$C$H=CH$_2$), 115.8 (−CH$_2$−CH=$C$H$_2$), 45.7, 45.6 (−$C$H$_2$−CH=CH$_2$), 47.7, 47.5 ppm (C$H_2$); MS (EI): $m/z$ (%) 351 (M$^+$+1, 1.3), 350 (M$^+$, 4.8), 309 (M$^+$−41, 100).

## 4.3. Synthesis of bis-thiazol-2-imine **7–9**

A mixture of bis-thioureas **3a-d** (1.0 mmol), o-methoxy phenacyl bromide (**6**) (2.0 mmol, 454.0 mg) and NaOAc (2.0 mmol) were sonicated in ethanol (20 ml) for 10 min at 80°C. The reaction was controlled by TLC and continued until the starting materials completely disappeared, then left to cool to room temperature. The formed solid was filtered off, washed with chilled ethanol (2 × 5 ml), dried and recrystallized from the proper solvent, to afford the pure products **7–9** with quantitative yields.

### 4.3.1. (2Z,2′Z)-3,3′-((1R,4R)-Cyclohexane-1,4-diyl)bis(N-benzyl-4-(2-methoxyphenyl)thiazol-2(3H)-imine) **7a**

Brown crystals, yield 98%, m.p. 311–314°C; IR (KBr): $\nu$/cm$^{-1}$ 1615 (C=N, C=C); $^1$H NMR (400 MHz, CDCl$_3$): $\delta$ 7.13–7.04 (m, 18H, Ar-H), 6.87 (s, 2H, triazole-H), 3.81 (s, 6H, OC$H_3$), 3.28–3.26 (m, 2H, Cyclohexane-H), 2.86 (s, 4H, C$H_2$), 1.96–1.95 (m, 4H, Cyclohexane-H), 1.74–1.69 ppm (m, 4H, Cyclohexane-H); $^{13}$C NMR (100 MHz, CDCl$_3$): $\delta$ 157.6 (Thiazole-C2), 144.9 (Thiazole-C4), 156.7, 138.3, 128.9, 127.6, 126.5, 125.8, 123.7, 122.4, 115.9, 107.4 (Ar-C), 108.3 (Thiazole-C5), 55.6 (OC$H_3$), 52.1 (Cyclohexane-C), 49.7 (C$H_2$) 31.0 ppm (Cyclohexane-C); MS (EI): $m/z$ (%) 673 (M$^+$+1, 2.4), 672 (M$^+$, 8.3), 581 (M$^+$−91, 100).

### 4.3.2. (2Z,2′Z)-3,3′-((1R,4R)-Cyclohexane-1,4-diyl)bis(N-allyl-4-(2-methoxyphenyl)thiazol-2(3H)-imine) **7b**

Brown crystals, yield 99%, m.p. 289–292°C; IR (KBr): $\nu$/cm$^{-1}$ 1604, 1598 (C=N, C=C); $^1$H NMR (400 MHz, DMSO-$d_6$): $\delta$ 7.28–7.23 (m, 8H, Ar-H), 6.92 (s, 2H, triazole-H), 5.82–5.73 (m, 2H, −CH$_2$−C$H$=CH$_2$), 5.11–5.05 (m, 4H, −CH$_2$−CH=C$H_2$), 4.19–4.17 (m, 4H, −C$H_2$−CH=CH$_2$), 3.79 (s, 6H, OC$H_3$), 3.18–3.13 (m, 2H, Cyclohexane-H), 1.95–1.93 (m, 4H, Cyclohexane-H), 1.27–1.21 ppm (m, 4H, Cyclohexane-H); $^{13}$C NMR (100 MHz, DMSO-$d_6$): $\delta$ 158.2 (Thiazole-C2), 145.7 (Thiazole-C4), 108.4 (Thiazole-C5), 157.5, 128.8, 122.4, 121.7, 114.7, 108.0 (Ar-C), 135.7 (−CH$_2$−$C$H=CH$_2$), 115.9 (−CH$_2$−CH=$C$H$_2$), 55.5 (OC$H_3$), 52.0 (Cyclohexane-C), 45.8 (−$C$H$_2$−CH=CH$_2$), 31.0 ppm (Cyclohexane-C); MS (EI): $m/z$ (%) 573 (M$^+$+1, 0.7), 572 (M$^+$, 3.9), 531 (M$^+$−41, 100).

### 4.3.3. (2Z,2′Z)-N,N′-(1,4-Phenylenebis(methylene))bis(3-benzyl-4-(2-methoxyphenyl)thiazol-2(3H)-imine) **8a**, (2Z,2′Z)-3,3′-(1,4-Phenylenebis(methylene))bis(N-benzyl-4-(2-methoxyphenyl)thiazol-2(3H)-imine) **9a**

The crude reaction mixture comprising both the non-isolable regioisomeric products was purified by column chromatography (92% DCM/MeOH) to afford the mixture of regioisomers **8a** and **9a** in the ratio of (56 : 44); Rf=0.39. Brown crystals, yield 96%, m.p. 341–343°C; IR (KBr): $\nu$/cm$^{-1}$ 1608, 1593 (C=N, C=C); $^1$H NMR (400 MHz, DMSO-$d_6$): $\delta$ 7.29–7.18 (m, Ar-H), 6.78–6.77 (m, triazole-H), 4.98–4.95 (m, C$H_2$), 4.65–4.61 (m, C$H_2$), 3.76 (d, 2 OC$H_3$); $^{13}$C NMR (100 MHz, DMSO-$d_6$): $\delta$ 158.0

(Thiazole-C2), 144.3 (Thiazole-C4), 108.0 (Thiazole-C5), 157.9, 138.7, 136.3, 134.8, 128.8, 128.3, 126.4, 125.5, 122.9, 120.3, 114.3, 107.3 (Ar-C), 56.9 (CH$_2$), 55.7, 55.2 (OCH$_3$), 48.4 ppm (CH$_2$); MS (EI): m/z (%) 696 (M$^+$+2, 1.3), 695 (M$^+$+1, 2.7), 694 (M$^+$, 100).

### 4.3.4. (2Z,2′Z)-N,N′-(1,4-Phenylenebis(methylene))bis(3-allyl-4-(2-methoxyphenyl)thiazol-2(3H)-imine) 8b, (2Z,2′Z)-3,3′-(1,4-Phenylenebis(methylene))bis(N-allyl-4-(2-methoxyphenyl)thiazol-2(3H)-imine) 9b

The crude reaction mixture comprising both the inseparable regioisomeric products was purified by column chromatography (94% DCM/MeOH) to afford the mixture of regioisomers **8b** and **9b** in the ratio of (52 : 48); Rf = 0.43. Brown crystals, yield 95%, m.p. 299–302°C decomposes; IR (KBr): ν/cm$^{-1}$ 1613, 1594 (C=N, C=C); $^1$H NMR (400 MHz, DMSO-$d_6$): δ 7.36–7.24 (m, Ar-H), 6.80 (d, triazole-H), 5.90 (m, −CH$_2$−CH=CH$_2$), 5.17–5.15 (m, −CH$_2$−CH=CH$_2$), 4.94–4.91 (d, 2 CH$_2$), 4.63–4.60 (d, 2 CH$_2$), 4.40–4.37 (m, CH$_2$−CH=CH$_2$), 3.79 (s, 2 OCH$_3$); $^{13}$C NMR (100 MHz, DMSO-$d_6$): δ 158.4 (Thiazole-C2), 144.6 (Thiazole-C4), 107.9 (Thiazole-C5), 158.1, 136.1, 134.7, 129.0, 128.7, 123.3, 121.8, 114.6, 107.5 (Ar-C), 135.7 (−CH$_2$−CH=CH$_2$), 115.5 (−CH$_2$−CH=CH$_2$), 45.7 (−CH$_2$−CH=CH$_2$), 56.7 (CH$_2$), 55.6, 55.3 (OCH$_3$), 48.7 ppm (CH$_2$); MS (EI): m/z (%) 595 (M$^+$+1, 23.6), 594 (M$^+$, 100).

## 4.4. Synthesis of derivatives 10a–d and 11a–d

A solution of bis-thioureas **3a–d** (1.0 mmol), malonic acid (2.2 mmol) in acetyl chloride (6.0 mmol) or oxalyl chloride (2.2 mmol), was sonicated at 40°C for 7 min. The solid product so obtained was filtered, washed with water (3 × 5 ml) then chilled ethanol (2 × 5 ml), and recrystallized from dioxane.

### 4.4.1. 3,3′-((1R,4R)-Cyclohexane-1,4-diyl)bis(1-benzyl-2-thioxodihydropyrimidine-4,6(1H,5H)-dione) 10a

Yellow crystals, yield 99%, m.p. 280–281°C; IR (KBr): ν/cm$^{-1}$ 1678, 1663 (C=O), 1350 (C=S); $^1$H NMR (400 MHz, DMSO-$d_6$): δ 7.33–7.28 (m, 10H, Ar-H), 4.92 (s, 4H, N−CH$_2$), 3.93 (br, 2H, Cyclohexane-H), 3.64 (s, 4H, CH$_2$), 1.81–1.79 (m, 4H, Cyclohexane-H), 1.56 (m, 4H, Cyclohexane-H); $^{13}$C NMR (100 MHz, DMSO-$d_6$): δ 171.8 (C=S), 166.8 (C=O), 166.5 (C=O), 136.0, 134.7, 128.7, 127.5 (Ar-C), 52.6 (Cyclohexane-C), 51.9 (N−CH$_2$) 41.7 (CH$_2$), 31.2 ppm (Cyclohexane-C); MS (EI): m/z (%) 549 (M$^+$+1, 8.6), 548 (M$^+$, 100).

### 4.4.2. 3,3′-((1R,4R)-Cyclohexane-1,4-diyl)bis(1-allyl-2-thioxodihydropyrimidine-4,6(1H,5H)-dione) 10b

Yellow crystals, yield 97%, m.p. 272–274°C; IR (KBr): ν/cm$^{-1}$ 1685, 1673 (C=O), 1348 (C=S); $^1$H NMR (400 MHz, DMSO-$d_6$): δ 5.90–5.77 (m, 2H, −CH$_2$−CH=CH$_2$), 5.16–5.04 (m, 4H, −CH$_2$−CH=CH$_2$), 4.03 (br, 4H, −CH$_2$−CH=CH$_2$), 3.91 (br, 2H, Cyclohexane-H), 3.56 (s, 4H, CH$_2$), 1.92–1.90 (m, 4H, Cyclohexane-H), 1.25–1.19 (m, 4H, Cyclohexane-H); $^{13}$C NMR (100 MHz, DMSO-$d_6$): δ 171.9 (C=S), 166.7 (C=O), 166.3 (C=O), 135.2 (−CH$_2$−CH=CH$_2$), 115.9 (−CH$_2$−CH=CH$_2$), 52.3 (Cyclohexane-C), 46.2 (−CH$_2$−CH=CH$_2$), 41.5 (CH$_2$), 31.0 ppm (Cyclohexane-C); MS (EI): m/z (%) 449 (M$^+$+1, 1.3), 448 (M$^+$, 100).

### 4.4.3. 3,3′-(1,4-Phenylenebis(methylene))bis(1-benzyl-2-thioxodihydropyrimidine-4,6(1H,5H)-dione) 10c

Yellow crystals, yield 99%, m.p. 309–311°C; IR (KBr): ν/cm$^{-1}$ 1668, 1660 (C=O), 1349 (C=S); $^1$H NMR (400 MHz, DMSO-$d_6$): δ 7.35–7.21 (m, 14H, Ar-H), 4.65 (d, 8H, 2 N−CH$_2$), 3.56 ppm (s, 4H, CH$_2$); $^{13}$C NMR (100 MHz, DMSO-$d_6$): δ 172.4 (C=S), 165.6 (C=O), 165.4 (C=O), 135.7, 135.2, 128.5, 128.2, 127.5, 126.8 (Ar-C) 51.5, 51.3, 41.1 ppm (CH$_2$); MS (EI): m/z (%) 571 (M$^+$+1, 1.3), 570 (M$^+$, 12.6), 479 (M$^+$−91, 100).

### 4.4.4. 3,3′-(1,4-Phenylenebis(methylene))bis(1-allyl-2-thioxodihydropyrimidine-4,6(1H,5H)-dione) 10d

Yellow crystals, yield 98%, m.p. 292–293°C; IR (KBr): ν/cm$^{-1}$ 1684, 1675 (C=O), 1390 (C=S); $^1$H NMR (400 MHz, DMSO-$d_6$): δ 7.27 (s, 4H, Ar-H), 5.89–5.81 (m, 2H, −CH$_2$−CH=CH$_2$), 5.17–5.05 (m, 4H, −CH$_2$−CH=CH$_2$), 4.64–4.62 (m, 4H, −CH$_2$−CH=CH$_2$), 4.06 (s, 4H, N−CH$_2$), 3.56 ppm (s, 4H, CH$_2$); $^{13}$C NMR (100 MHz, DMSO-$d_6$): δ 172.2 (C=S), 165.7 (C=O), 165.4 (C=O), 135.3, 128.1 (Ar-C), 135.0 (−CH$_2$−CH=CH$_2$), 115.6 (−CH$_2$−CH=CH$_2$), 51.7 (CH$_2$), 46.4 (−CH$_2$−CH=CH$_2$), 41.0 ppm (CH$_2$); MS (EI): m/z (%) 471 (M$^+$+1, 1.2), 470 (M$^+$, 100).

### 4.4.5. 3,3′-((1R,4R)-Cyclohexane-1,4-diyl)bis(1-benzyl-2-thioxoimidazolidine-4,5-dione) 11a

Yellow crystals, yield 96%, m.p. 282–283°C; IR (KBr): $\nu$/cm$^{-1}$ 1696, 1687 (C=O), 1358 (C=S); $^1$H NMR (400 MHz, DMSO-$d_6$): $\delta$ 7.35–7.21 (m, 10H, Ar-H), 4.77 (s, 4H, C$H_2$), 3.92 (br, 2H, Cyclohexane-H), 1.95–1.92 (m, 4H, Cyclohexane-H), 1.28–1.21 (m, 4H, Cyclohexane-H); $^{13}$C NMR (100 MHz, DMSO-$d_6$): $\delta$ 167.9 (C=S), 161.7 (C=O), 160.9 (C=O), 135.5, 128.5, 127.9, 127.0 (Ar–C), 52.5 (Cyclohexane-C), 51.5 (CH$_2$), 30.8 ppm (Cyclohexane-C); MS (EI): $m/z$ (%) 521 (M$^+$+1, 6.3), 520 (M$^+$, 9.6), 429 (M$^+$−91, 100).

### 4.4.6. 3,3′-((1R,4R)-Cyclohexane-1,4-diyl)bis(1-allyl-2-thioxoimidazolidine-4,5-dione) 11b

Yellow crystals, yield 97%, m.p. 277–279°C; IR (KBr): $\nu$/cm$^{-1}$ 1695, 1689 (C=O), 1385 (C=S); $^1$H NMR (400 MHz, DMSO-$d_6$): $\delta$ 5.89–5.81 (m, 2H, −CH$_2$−C$H$=CH$_2$), 5.17–5.04 (m, 4H, −CH$_2$−CH=C$H_2$), 4.05 (br, 4H, −C$H_2$−CH=CH$_2$), 3.80 (br, 2H, Cyclohexane-H), 1.90 (br, 4H, Cyclohexane-H), 1.22 (br, 4H, Cyclohexane-H); $^{13}$C NMR (100 MHz, DMSO-$d_6$): $\delta$ 167.7 (C=S), 161.5 (C=O), 160.7 (C=O), 135.3 (−CH$_2$−CH=CH$_2$), 115.9 (−CH$_2$−CH=CH$_2$), 46.7 (−CH$_2$−CH=CH$_2$), 51.8 (Cyclohexane-C), 31.3 ppm (Cyclohexane-C); MS (EI): $m/z$ (%) 421 (M$^+$+1, 3.0), 420 (M$^+$, 100).

### 4.4.7. 3,3′-(1,4-Phenylenebis(methylene))bis(1-benzyl-2-thioxoimidazolidine-4,5-dione) 11c

Yellow crystals, yield 93%, m.p. 317–320°C; IR (KBr): $\nu$/cm$^{-1}$ 1705, 1698 (C=O), 1390 (C=S); $^1$H NMR (400 MHz, DMSO-$d_6$): $\delta$ 7.35–7.21 (m, 14H, Ar-H), 4.66 (s, 4H, C$H_2$), 4.64 (s, 4H, C$H_2$); $^{13}$C NMR (100 MHz, DMSO-$d_6$): $\delta$ 167.4 (C=S), 161.4 (C=O), 161.1 (C=O), 136.0, 134.9, 128.8, 128.0, 127.1, 126.8 (Ar-C) 51.7, 51.5 ppm (CH$_2$); MS (EI): $m/z$ (%) 543 (M$^+$+1, 7.7), 542 (M$^+$, 100).

### 4.4.8. 3,3′-(1,4-Phenylenebis(methylene))bis(1-allyl-2-thioxoimidazolidine-4,5-dione) 11d

Yellow crystals, yield 96%, m.p. 299–302°C; IR (KBr): $\nu$/cm$^{-1}$ 1711, 1704 (C=O), 1395 (C=S); $^1$H NMR (400 MHz, DMSO-$d_6$): $\delta$ 7.23 (s, 4H, Ar-H), 5.90–5.80 (m, 2H, −CH$_2$−C$H$=CH$_2$), 5.18–5.03 (m, 4H, −CH$_2$−CH=C$H_2$), 4.65–4.63 (m, 4H, −C$H_2$−CH=CH$_2$), 4.05 ppm (s, 4H, C$H_2$); $^{13}$C NMR (100 MHz, DMSO-$d_6$): $\delta$ 166.4 (C=S), 161.3 (C=O), 161.2 (C=O), 135.5 (−CH$_2$−CH=CH$_2$), 134.6, 128.0 (Ar-C), 115.8 (−CH$_2$−CH=CH$_2$), 51.9 (CH$_2$), 46.5 ppm (−CH$_2$−CH=CH$_2$); MS (EI): $m/z$ (%) 443 (M$^+$+1, 5.9), 442 (M$^+$, 100).

## 4.5. Synthesis of bis-2,1-benzisoxazole derivatives 13a–d

A mixture of bis-thiobarbituric acids 10a–d (1.0 mmol) and o-nitrobenzaldehyde (2.0 mmol) in water (10 ml) was sonicated at 80°C for 5 min. Then, water was removed under reduced pressure and the remaining residue was suspended in ethanol (10 ml), to which, tin chloride dihydrate (4.0 mmol) and concentrated hydrochloric acid (2 ml) was added. The resulting mixture was then sonicated for 10 min at 80°C. The obtained hot mixture was filtered off. The solid was washed with diethyl ether (2 × 5 ml) and recrystallized form dioxane containing few drops of DMF to afford bis-2,1-benzisoxazoles.

### 4.5.1. (5Z,5′Z)-3,3′-((1R,4R)-Cyclohexane-1,4-diyl)bis(5-(benzo[c]isoxazol-3(1H)-ylidene)-1-benzyl-2-thioxodihydropyrimidine-4,6(1H,5H)-dione) 13a

Brown crystals, yield 92%, m.p. 298–301°C decomposes; IR (KBr): $\nu$/cm$^{-1}$ 3220–3053 (br, NH), 1686 (C=O), 1617, 1590 (C=C), 1387 (C=S); $^1$H NMR (400 MHz, DMSO-$d_6$): $\delta$ 7.89 (d, $J$ = 8.7 Hz, 2H, Ar-H), 7.35–7.23 (m, 14H, Ar-H), 6.77 (d, $J$ = 8.8 Hz, 2H, Ar-H), 4.82 (s, 4H, C$H_2$), 3.93 (br, 2H, Cyclohexane-H), 1.92–1.90 (m, 4H, Cyclohexane-H), 1.25–1.19 (m, 4H, Cyclohexane-H); $^{13}$C NMR (100 MHz, DMSO-$d_6$): $\delta$ 178.6 (C=S), 167.3 (C3′), 161.9 (C=O), 161.2 (C=O), 156.5, 136.1, 129.0, 128.4, 127.2, 125.6, 123.9, 119.0, 114.5, 113.3 (Ar-C), 86.8 (C5), 52.8 (Cyclohexane-C), 51.0 (CH$_2$), 30.3 ppm (Cyclohexane-C); MS (EI): $m/z$ (%) 783 (M$^+$+1, 0.6), 782 (M$^+$, 1.6), 691(M$^+$−91, 100).

### 4.5.2. (5Z,5′Z)-3,3′-((1R,4R)-Cyclohexane-1,4-diyl)bis(1-allyl-5-(benzo[c]isoxazol-3(1H)-ylidene)-2-thioxodihydropyrimidine-4,6(1H,5H)-dione) 13b

Brown crystals, yield 89%, m.p. 287–288°C decomposes; IR (KBr): $\nu$/cm$^{-1}$ 3262–3011 (br, NH), 1677 (C=O), 1619, 1584 (C=C), 1345 (C=S); $^1$H NMR (400 MHz, DMSO-$d_6$): $\delta$ 7.80 (d, $J$ = .9 Hz, 2H, Ar-H), 7.29–7.18 (m, 4H, Ar-H), 6.77 (d, $J$ = 8.8 Hz, 2H, Ar-H), 5.89–5.80 (m, 2H, −CH$_2$−C$H$=CH$_2$),

5.17–5.05 (m, 4H, –CH$_2$–CH=C*H*$_2$), 4.63 (m, 4H, –C*H*$_2$–CH=CH$_2$), 3.49 (br, 2H, Cyclohexane-H), 1.76–1.74 (m, 4H, Cyclohexane-H), 1.45–1.39 (m, 4H, Cyclohexane-H); $^{13}$C NMR (100 MHz, DMSO-$d_6$): δ 178.5 (C=S), 167.9 (C3′), 161.2 (C=O), 160.8 (C=O), 156.8, 130.6, 125.6, 120.0, 114.4, 113.8 (Ar-C), 135.1 (–CH$_2$–CH=CH$_2$), 115.6 (–CH$_2$–CH=CH$_2$), 86.2 (C5), 51.9 (Cyclohexane-C), 46.9 (–CH$_2$–CH=CH$_2$), 31.0 ppm (Cyclohexane-C); MS (EI): $m/z$ (%) 683 (M$^+$+1, 3.5), 682 (M$^+$, 100).

### 4.5.3. (5Z,5′Z)-3,3′-(1,4-Phenylenebis(methylene))bis(5-(benzo[c]isoxazol-3(1H)-ylidene)-1-benzyl-2-thioxodihydropyrimidine-4,6(1H,5H)-dione) 13c

Red crystals, yield 93%, m.p. 311–313°C decomposes; IR (KBr): $\nu$/cm$^{-1}$ 3206–2863 (br, NH), 1669 (C=O), 1625, 1603 (C=C), 1344 (C=S); $^1$H NMR (400 MHz, DMSO-$d_6$): δ 7.98 (d, $J$ = 8.7 Hz, 2H, Ar-H), 7.32–7.22 (m, 18H, Ar-H), 6.78 (d, $J$ = 8.7 Hz, 2H, Ar-H), 4.78 (s, 8H, C*H*$_2$); $^{13}$C NMR (100 MHz, DMSO-$d_6$): δ 179.0 (C=S), 166.8 (C3′), 162.0 (C=O), 161.4 (C=O), 155.9, 136.0, 131.3, 128.8, 128.2, 127.6, 127.2, 126.9, 126.2, 119.6, 114.8, 113.2 (Ar-C), 86.2 (C5), 51.9, 51.4 ppm (CH$_2$); MS (EI): $m/z$ (%) 805 (M$^+$+1, 1.4), 804 (M$^+$, 100).

### 4.5.4. (5Z,5′Z)-3,3′-(1,4-Phenylenebis(methylene))bis(1-allyl-5-(benzo[c]isoxazol-3(1H)-ylidene)-2-thioxodihydropyrimidine-4,6(1H,5H)-dione) 13d

Yellow crystals, yield 95%, m.p. 297–299°C; IR (KBr): $\nu$/cm$^{-1}$ 3341–3069 (br, NH), 1683 (C=O), 1624, 1598 (C=C), 1369 (C=S); $^1$H NMR (400 MHz, DMSO-$d_6$): δ 7.89 (br, 2H, Ar-H), 7.35–7.23 (m, 8H, Ar-H), 6.80 (d, $J$ = 8.8 Hz, 2H, Ar-H), 5.89–5.80 (m, 2H, –CH$_2$–C*H*=CH$_2$), 4.99 (br, 4H, –CH$_2$–CH=C*H*$_2$), 4.56 (br, 4H, –C*H*$_2$–CH=CH$_2$), 4.01 ppm (m, 4H, C*H*$_2$); $^{13}$C NMR (100 MHz, DMSO-$d_6$): δ 179.1 (C=S), 167.4 (C3′), 161.5 (C=O), 161.1 (C=O), 155.9, 134.6, 130.7, 128.3, 126.1, 119.5, 114.2, 113.6 (Ar-C), 135.7 (–CH$_2$–CH=CH$_2$), 115.9 (–CH$_2$–CH=CH$_2$), 51.7 (CH$_2$), 86.2 (C5), 46.4 ppm (–CH$_2$–CH=CH$_2$); MS (EI): $m/z$ (%) 705 (M$^+$+1, 2.8), 704 (M$^+$, 100).

Data accessibility. The datasets supporting this article have been uploaded as part of the electronic supplementary material.

Authors' contributions. W.A.A.A. designed and performed experimental part of the work, managed the research, performed data analysis and wrote the manuscript. H.M.I. assisted with designing, performing experimental part, data analysis, writing and revising of the manuscript.

Competing interests. We have no competing interests.

Funding. This study was financially supported by the Jouf University, project no. 634/39.

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
