## [Reviewer comments · Royal Society Open Science]

Review History

RSOS-181963.R0 (Original submission)

Review form: Reviewer 1 (Yuriy G. Shermolovich)

Is the manuscript scientifically sound in its present form?

Yes

Are the interpretations and conclusions justified by the results?

Yes

Is the language acceptable?

Yes

Is it clear how to access all supporting data?

Yes

Do you have any ethical concerns with this paper?

No

Have you any concerns about statistical analyses in this paper?

No

Recommendation?

Accept as is

Comments to the Author(s)

The authors synthesized new compounds using known approaches to the preparation of the heterocycles indicated by them. The novelty of the work lies in the application of the sonication method, which allows to significantly increase the yield of the target compounds. The work will certainly be of interest to specialists working in the field of medical chemistry. As a wish, I would like to suggest the following to the authors: having such a *yield* of new stable crystalline compounds, it would be desirable to study the structure of at least some of them using X-ray diffraction method.

Review form: Reviewer 2

Is the manuscript scientifically sound in its present form?

Yes

Are the interpretations and conclusions justified by the results?

Yes

Is the language acceptable?

Yes

Is it clear how to access all supporting data?

Yes

Do you have any ethical concerns with this paper?

No

Have you any concerns about statistical analyses in this paper?

No

Recommendation?

Major revision is needed (please make suggestions in comments)

Comments to the Author(s)

The authors report a new one-pot protocol for sono-synthesis of bis-[1,2,4]-triazol-3-yl amines and bis-2-iminothiazolines from bis-1,3-disubstituted thioureas and an efficient methodology to sono-synthesize of bis-isoxazolthiobarbiturates and bis-imidazoles from bis-1,3-disubstituted thioureas as well. However, some statements are obviously exaggerated such as "the other advantages of this novel sonochemical protocol are no tedious work-up and varied functional group tolerance".

(1) As described in the Experimental section, the synthesis and work-up process of new compounds seemed to be basically the same as normal procedure except for an ultrasonic process. The improvements of the ultrasonic process should be discussed in more detail.

(2) The authors claimed that “The products regioselectivity correlate well with the pKas of the parent amines”. However, there are only two kinds of R group and two kinds of R' group be introduced as is shown in Table 2. Thus, more functional groups should be explored if possible.

Review form: Reviewer 3

Is the manuscript scientifically sound in its present form?

Yes

Are the interpretations and conclusions justified by the results?

Yes

Is the language acceptable?

Yes

Is it clear how to access all supporting data?

Yes

Do you have any ethical concerns with this paper?

No

Have you any concerns about statistical analyses in this paper?

No

Recommendation?

Reject

Comments to the Author(s)

In this manuscript (RSOS-181963), Wael and coworker reported the synthesis of polysubstituted bis-1,2,4-triazoles, bis-2-iminothiazolines and bis-thiobarbiturates from bis-N,N-disubstituted thioureas under ultrasound irradiation. The influence of the pKas of the parent amines was discussed to explain the regioselectivity of products. Further derivatization of the resulting products also provided some new heterocycles. While this work is correctly done, one major problem is that the same authors have previously reported the similar reactions (ref. 40, RSC Adv., 2018, 8, 10516). The major novelty of this work has been covered in their previous report, making the present study a derivative work. Thus I regret not to support the publication of this work in RSOS. Other more specific journal such as Journal of Heterocyclic Chemistry is more suitable for this work.

Additional comments:

1. The substituents in Scheme 1 are not correct. “benzyl” and “allyl” should be “Ph” and “CH=CH₂” respectively.
2. The data in Table 3 should be revised. Two significant digit is recommended for the product yield and AE (e.g. entry 1: “98.13%” should be “98%”).
3. I am not quite sure about the authors’ names. In the author list, the last name of the corresponding author is “Arafa”, while in the cited references the last name is “Wael” (also see other papers published by the same authors). The authors should check this information.

Decision letter (RSOS-181963.R0)

05-Apr-2019

Dear Professor Arafa:

Title: Sustainable synthesis of bis-1,2,4-triazoles, bis-2-iminothiazolines and bis-thiobarbiturates utilizing bis-N,N-disubstituted thioureas
Manuscript ID: RSOS-181963

The editor assigned to your manuscript has now received comments from reviewers. We would like you to revise your paper in accordance with the referee and Subject Editor suggestions which can be found below (not including confidential reports to the Editor). Please note this decision does not guarantee eventual acceptance.

Please submit your revised paper before 28-Apr-2019. Please note that the revision deadline will expire at 00.00am on this date. If we do not hear from you within this time then it will be assumed that the paper has been withdrawn. In exceptional circumstances, extensions may be possible if agreed with the Editorial Office in advance. We do not allow multiple rounds of revision so we urge you to make every effort to fully address all of the comments at this stage. If deemed necessary by the Editors, your manuscript will be sent back to one or more of the original reviewers for assessment. If the original reviewers are not available we may invite new reviewers.

Please also include the following statements alongside the other end statements. As we cannot publish your manuscript without these end statements included, if you feel that a given heading is not relevant to your paper, please nevertheless include the heading and explicitly state that it is not relevant to your work.

- Acknowledgements

Yours sincerely,
Dr Laura Smith

Publishing Editor, Journals

On behalf of the Subject Editor Professor Anthony Stace and the Associate Editor Professor John Moses.

RSC Associate Editor:
Comments to the Author:
(There are no comments.)

RSC Subject Editor:
Comments to the Author:
(There are no comments.)

Reviewers' Comments to Author:
Reviewer: 1

Comments to the Author(s)

The authors synthesized new compounds using known approaches to the preparation of the heterocycles indicated by them. The novelty of the work lies in the application of the sonication method, which allows to significantly increase the yield of the target compounds. The work will certainly be of interest to specialists working in the field of medical chemistry. As a wish, I would like to suggest the following to the authors: having such a *yield* of new stable crystalline compounds, it would be desirable to study the structure of at least some of them using X-ray diffraction method.

Reviewer: 2

Comments to the Author(s)

The authors report a new one-pot protocol for sono-synthesis of bis-[1,2,4]-triazol-3-yl amines and bis-2-iminothiazolines from bis-1,3-disubstituted thioureas and an efficient methodology to sono-synthesize of bis-isoxazolthiobarbiturates and bis-imidazoles from bis-1,3-disubstituted thioureas as well. However, some statements are obviously exaggerated such as "the other advantages of this novel sonochemical protocol are no tedious work-up and varied functional group tolerance".

(1) As described in the Experimental section, the synthesis and work-up process of new compounds seemed to be basically the same as normal procedure except for an ultrasonic process. The improvements of the ultrasonic process should be discussed in more detail.
(2) The authors claimed that "The products regioselectivity correlate well with the pK_as of the parent amines". However, there are only two kinds of R group and two kinds of R' group be introduced as is shown in Table 2. Thus, more functional groups should be explored if possible.

Reviewer: 3

Comments to the Author(s)

In this manuscript (RSOS-181963), Wael and coworker reported the synthesis of polysubstituted bis-1,2,4-triazoles, bis-2-iminothiazolines and bis-thiobarbiturates from bis-N,N-disubstituted thioureas under ultrasound irradiation. The influence of the pKas of the parent amines was discussed to explain the regioselectivity of products. Further derivatization of the resulting products also provided some new heterocycles. While this work is correctly done, one major problem is that the same authors have previously reported the similar reactions (ref. 40, RSC Adv., 2018, 8, 10516). The major novelty of this work has been covered in their previous report, making the present study a derivative work. Thus I regret not to support the publication of this work in RSOS. Other more specific journal such as Journal of Heterocyclic Chemistry is more suitable for this work.

Additional comments:

1. The substituents in Scheme 1 are not correct. "benzyl" and "allyl" should be "Ph" and "CH=CH₂" respectively.

2. The data in Table 3 should be revised. Two significant digit is recommended for the product yield and AE (e.g. entry 1: "98.13%" should be "98%").

3. I am not quite sure about the authors' names. In the author list, the last name of the corresponding author is "Arafa", while in the cited references the last name is "Wael" (also see other papers published by the same authors). The authors should check this information.

Author's Response to Decision Letter for (RSOS-181963.R0)

See Appendix A.

RSOS-181963.R1 (Revision)

Review form: Reviewer 2

Is the manuscript scientifically sound in its present form?

Yes

Are the interpretations and conclusions justified by the results?

Yes

Is the language acceptable?

Yes

Is it clear how to access all supporting data?

Yes

Do you have any ethical concerns with this paper?

No

Have you any concerns about statistical analyses in this paper?

No

Recommendation?

Accept as is

Comments to the Author(s)

Accept as is

Decision letter (RSOS-181963.R1)

22-May-2019

Dear Professor Arafa:

Title: Sustainable synthesis of bis-1,2,4-triazoles, bis-2-iminothiazolines and bis-thiobarbiturates utilizing bis-N,N-disubstituted thioureas
Manuscript ID: RSOS-181963.R1

It is a pleasure to accept your manuscript in its current form for publication in Royal Society Open Science. The chemistry content of Royal Society Open Science is published in collaboration with the Royal Society of Chemistry.

The comments of the reviewer(s) who reviewed your manuscript are included at the end of this email. I apologise that this took longer than usual.

On behalf of the Subject Editor Professor Anthony Stace and the Associate Editor Professor John Moses.

RSC Associate Editor:
Comments to the Author:
(There are no comments.)

RSC Subject Editor:
Comments to the Author:
(There are no comments.)

Reviewer(s)' Comments to Author:
Reviewer: 2

Comments to the Author(s)
Accept as is

Appendix A

Entry	Reviewer	Comments	Responses
1.	Editor	Please also include the following statements alongside the other end statements. "Acknowledgements"	This comment has been satisfied (highlighted in yellow).
2.	R1	Having such a urye of new stable crystalline compounds, it would be desirable to study the structure of at least some of them using X-ray diffraction method.	All attempts that have been made to grow up a single crystal suitable for X-ray diffraction have been failed.
3.	R2	However, some statements are obviously exaggerated such as "the other advantages of this novel sonochemical protocol are no tedious work-up and varied functional group tolerance".	This statement has been replaced by: "The reactions proceed smoothly affording target products in excellent yields." (highlighted in yellow).
4.		As described in the Experimental section, the synthesis and work-up process of new compounds seemed to be basically the same as normal procedure except for an ultrasonic process. The improvements of the ultrasonic process should be discussed in more detail.	In Result and discussion part, a paragraph relating to the role of ultrasonic irradiation has been inserted and references No. 45-47 have been cited. (highlighted in yellow color).
5.		The authors claimed that "The products regioselectivity correlate well with the pKas of the parent amines". However, there are only two kinds of R group and two kinds of R' group be introduced as is shown in Table 2. Thus, more functional groups should be explored if possible.	For synthesizing of bis-[1,2,4]-triazol-3-yl amines and bis-2-iminothiazolines we started with four derivatives (3a-d) . According to their pKas, the reaction yielded different regioselective products (4-8). In my opinion, that is enough (4 examples) and the synthesis of more derivatives will not give more information about the reactions. The other critical points are the synthesis of more derivatives needs more time and more different starting materials which are not available in our laboratory at this time.

6.	R3	While this work is correctly done, one major problem is that the same authors have previously reported the similar reactions (ref. 40, RSC Adv., 2018, 8, 10516).	Thank you for your comment. We believe that our previously published paper (RSC Adv., 2018, 8, 10516), in my opinion, is not a major problem due to this is a continuation to our research line. Also, the molecules that synthesized in the first paper are structurally different by those reported in the current paper.
7.		The substituents in Scheme 1 are not correct. "benzyl" and "allyl" should be "Ph" and "CH=CH ₂ " respectively.	This comment has been satisfied.
8.		The data in Table 3 should be revised. Two significant digit is recommended for the product yield and AE (e.g. entry 1: "98.13%" should be "98%").	The comment has been satisfied.
9.		I am not quite sure about the authors' names. In the author list, the last name of the corresponding author is "Arafa", while in the cited references the last name is "Wael" (also see other papers published by the same authors). The authors should check this information.	This comment has been satisfied (highlighted in yellow).